# PPI-based screening of hub genes related to sepsis migration/pyroptosis and immune infiltration analysis

Xu Ma[1]⊙, Jianhao Wang[2,3]⊙, Kun Han[2], Jinshuai Lu[3]*

**1** Department of General Surgery, Bai niao hu Hospital of Xinjiang Uygur Autonomous Region People's Hospital, Xinjiang Hospital of the Second Affiliated Hospital of Xi'an Jiao tong University, Urumqi, China, **2** Graduate School, Xinjiang Medical University, Urumqi, China, **3** Second Department of Emergency Center, People's Hospital of Xinjiang Uygur Autonomous Region, Urumqi, China

⊙ These authors contributed equally to this work.
* maxv0329@163.com

## Abstract

### Background

Sepsis incidence is rising, but its pathogenesis remains unclear. This study aimed to identify therapeutic targets.

### Methods

Three GEO datasets (GSE154918, GSE32707, GSE54514) were analyzed after batch correction. Differentially expressed genes (DEGs) were identified, and those related to pyroptosis/migration were cross-analyzed. Functional enrichment (GO/KEGG) and interaction networks (PPI, mRNA-miRNA, mRNA-TF, mRNA-drug) were constructed. External validation used GSE57065, excluding non-significant genes. ROC analysis and immune infiltration were performed.

### Results

3566 DEGs (1715 up, 1851 down) were identified, with 23 linked to pyroptosis/migration. Enrichment analysis highlighted roles in cell adhesion, cytokine regulation, and pathways like IL-17, TNF, and Toll-like receptor signaling. mRNA-miRNA interactions for 12 key genes were predicted. After validation, eight key genes remained: *TLR2, SIRT1, PTGS2, MAPK14, IL18, ICAM1, CD274*, and *CASP3*. Immune infiltration revealed varied effects on MAPK14.

### Conclusion

Key gene alterations may serve as sepsis biomarkers, and miRNA dysregulation could play a critical role in sepsis pathophysiology.

**Data availability statement:** The datasets analyzed during the current study is available in the GEO repository. GSE154918(https://www.ncbi.nlm.nih.gov/geo/query/acc.cgi?acc=GSE154918), GSE32707(https://www.ncbi.nlm.nih.gov/geo/query/acc.cgi?acc=GSE32707), and GSE54514(https://www.ncbi.nlm.nih.gov/geo/query/acc.cgi?acc=GSE54514).

**Funding:** This study was supported by the "Tianshan Talents" medical and health high-level Talents Training Program of Xinjiang Uygur Autonomous Region Health Commission (Project number: TSYC202301B038).

**Competing interests:** The authors have declared that no competing interests exist.

## Introduction

Sepsis is defined as a potentially life-threatening systemic dysfunction caused by a dysregulated host response to infection. The progression of this condition can lead to septic shock and dysfunction or failure of vital organs. According to the clinical criteria established by the latest consensus conference (Sepsis-3, the Third International Consensus Definitions for Sepsis and Septic Shock), diagnosing sepsis requires an increase of at least 2 points in the Sequential Organ Failure Assessment (SOFA) score from baseline [1]. Critically ill patients, particularly those with severe comorbidities, are more prone to multiple organ failure [2]. Septic shock is characterized by persistent hypotension that does not respond to adequate fluid resuscitation, and it is associated with a mortality rate as high as 27%−36% in developed countries [3]. The primary mechanism underlying sepsis-induced organ dysfunction is currently believed to be impaired tissue perfusion and abnormal cellular energy metabolism resulting from persistent hemodynamic disturbances [4]. Due to delayed diagnosis and poor prognosis, sepsis remains a highly lethal condition. Although several biomarkers (e.g., C-reactive protein, procalcitonin) have been reported for clinical diagnosis and management of septic patients [5], the complex clinical manifestations of sepsis and the limited sensitivity and specificity of some biomarkers continue to pose challenges for early diagnosis and treatment. These issues urgently require further investigation.

In recent years, the role of non-coding RNAs (ncRNAs) in regulating biological processes such as cell proliferation, differentiation, apoptosis, and cell cycle control has been gradually untangled. A growing body of research has focused on utilizing these molecules as diagnostic markers or therapeutic agents [6,7]. Among the most extensively studied ncRNAs are microRNAs (miRNAs or miRs). miRNAs are small molecules consisting of 19−25 ribonucleotides that are not translated into proteins but regulate the expression of specific target genes at the post-transcriptional level [8]. Given their role in epigenetic regulation and their widespread presence in tissues, intracellularly, and extracellularly, miRNAs not only hold potential as serum biomarkers for sepsis but may also serve as specific markers for assessing organ involvement during the pathological process [9,10]. Cell migration and pyroptosis are key processes in the pathogenesis of sepsis. In sepsis, cell migration involves not only the movement of immune cells in response to infection but also the infiltration and mobilization of inflammatory cells. These processes are crucial for modulating the intensity, duration, and scope of the inflammatory response; however, dysregulated immune response is considered one of the most relevant pathological mechanisms in sepsis [11]. Pyroptosis is a form of programmed cell death mediated by the activation of caspases (e.g., caspase-1/4/5/11), accompanied by cleavage of Gasdermin D (GSDMD) and massive release of pro-inflammatory cytokines. While it can facilitate pathogen clearance, excessive activation may exacerbate inflammatory injury and multiple organ failure in sepsis. Studies have shown that the level of pyroptosis in peripheral blood mononuclear cells of septic patients is closely associated with disease severity and mortality [12–14].

Although both cell migration and pyroptosis play important roles in sepsis, research on their interactions and shared molecular mechanisms remains limited. This study employs bioinformatics approaches to integrate multiple Gene Expression Omnibus (GEO) datasets, systematically analyze differentially expressed genes in sepsis, and focus on the screening and functional exploration of genes related to cell migration and pyroptosis. Furthermore, mRNA–miRNA, mRNA–transcription factor, and mRNA–drug interaction networks are constructed to untangle key regulatory mechanisms and identify potential therapeutic targets, thereby providing new directions for early diagnosis and precision treatment of sepsis.

## 2. Materials and methods

### 2.1 Data download

Three sepsis expression profile datasets, GSE154918 [15] (control group = 40, sepsis group = 20; 45 cases including incomplete infection cases, septic shock cases, and follow-up cases were excluded), GSE32707 [16] (control group = 34, sepsis group = 58; 31 ARDS cases were excluded), and GSE54514 [17] (control group = 36, sepsis group = 127; all cases were included), were downloaded from the GEO database [18] using the R package GEOquery [19]. The species source for all three datasets was human, and the tissue source was blood. The sequencing platform used was the Illumina microarray platform, with platform identifiers GPL20301, GPL10558, and GPL6947, respectively. Platform annotation files for each dataset were downloaded to convert probe IDs to gene symbols. For genes represented by multiple probes, the average expression value was calculated and used (see Table 1). Given that all three datasets focus on sepsis and share consistent sample types, they possess a biological basis for combined analysis. To minimize the impact of batch effects and technical noise on subsequent analyses, the ComBat method from the sva package was applied to perform batch correction on the raw expression data. The ComBat algorithm effectively adjusts for systematic biases introduced by non-biological factors such as sequencing platforms, experimental batches, and sample processing, thereby enhancing comparability across datasets. The effectiveness of correction was evaluated using principal component analysis (PCA) and boxplots of gene expression distributions, which demonstrated improved consistency in expression patterns post-correction (Fig 2). All subsequent analyses, including differential expression analysis, functional enrichment, and network construction, were conducted using the batch-corrected integrated dataset to minimize the influence of technical artifacts on the study conclusions.

### 2.2 Acquisition of genes associated with cell migration and pyroptosis

Via GeneCards [23] the genes related to cell migration and pyroptosis were collected. The GeneCards database offers a comprehensive repository of information on human genes, from which we have retrieved 196 genes related to pyroptosis by employing the keyword "pyroptosis" in our search and setting a relevance score of 1. Similarly, searching with the word "cell migration" as a keyword and screening with the relevance score of 1 yielded 1, 942 cell migration-related genes. 83 genes related to cell migration and pyroptosis were obtained after intersection, as shown in S1 Table for specific information.

**Table 1. GEO data set information.**

|  | GSE154918 | GSE32707 | GSE54514 |
|---|---|---|---|
| Species | Homo sapiens | Homo sapiens | Homo sapiens |
| Tissue | whole blood | whole blood | whole blood |
| Samples in Sepsis group | 20 | 58 | 127 |
| Samples in Normal group | 40 | 34 | 36 |
| Platform | GPL20301 | GPL10558 | GPL6947 |
| Reference | [20] | [21] | [22] |

## 2.3 Integration and difference analysis of data sets

For the batch-corrected GSE154918, GSE32707, and GSE54514 datasets, we performed data synthesis using the R package sva [24], ultimately obtaining an integrated dataset comprising 205 cases and 110 controls (hereinafter referred to as the "combined dataset"). This dataset was normalized using the R package limma, followed by differential gene expression analysis comparing cases versus controls. Given that this study involves peripheral blood samples from sepsis patients, where gene expression changes are generally subtle, a preliminary screening strategy was adopted: a threshold of $|logFC| > 0$ and p.adjust $< 0.05$ was set for identifying differentially expressed genes (DEGs). This approach was designed to maintain high biological sensitivity at the initial stage, ensuring that potentially meaningful signals were not prematurely filtered out. All DEGs identified in this phase were subjected to subsequent, more rigorous multi-step validation—including protein–protein interaction network analysis and independent cohort validation—to ensure the robustness of the final hub gene selection.

## 2.4 Gene ontology (GO) and Kyoto Encyclopedia of Genes and Genomes (KEGG) enrichment analysis

GO [25]is a standardised methodology for comprehensive functional enrichment studies, encompassing the domains of molecular functions, cellular components, and biological processes. The KEGG [26] a comprehensive database, contains information on a range of subjects including biological pathways, genomes, drugs, and diseases. The R package cluster Profiler [27]was utilized for this analysis. In order to gain insight into the biological mechanisms underlying differential gene expression in contexts of pyroptosis and cellular migration, a GO and a KEGG enrichment analysis were carried out on the genes in question. The initial screening criteria were set at p.adjust $< 0.05$ (or p-value $< 0.05$) and false discovery rate (FDR) (q-value) $< 0.05$, which were established as the threshold for statistical significance. The correction method of p.adjust was Benjamini-Hochberg (BH). Finally, we used the R package Pathview [28] to perform visualization of the relevant pathway for the pathway (KEGG) enrichment analysis.

## 2.5 Gene Set Enrichment Analysis (GSEA)

The GSEA [29], is a statistical method used to determine the distribution patterns of a gene within a pre-defined gene set in a gene list, with the aim of assessing its contribution to a phenotype. The method involves sorting the gene list based on a phenotypic correlation and then evaluating each gene within this list for enrichment within the gene set. In the present study, the R-Cluster Profiler was employed to conduct a GSEA for all genes included in the Combined Datasets. The following criteria were employed in the GSEA analysis: In the initial phase, 2,020 genes were selected as the seeds, and the subsequent calculations were conducted 1,000 times using a threshold of 10 genes in each gene subset and a ceiling of 500 genes per subset. The BH correction method for p-values was utilized. A gene set for GSEA was obtained from the Molecular Signatures Database (MSigDB) [30]. This gene set was derived from the C2.all.v7.2.symbols.gmt dataset and was used to screen for significant enrichment. The threshold for meaningful enrichment was set at p.adjust $< 0.05$ and FDR (q-value) $< 0.25$.

## 2.6 Screening of protein-protein interaction network and hub gene

The PPI network, is a network comprising proteins and their interactions with each other. The STRING Database [31] (Search Tool for the Retrieval of Interacting Genes/Proteins) is a repository of interactions between known and predicted proteins, as well as other biological data. In the present study, the STRING database was employed to identify genes exhibiting differential expression based on phenotypic correlation. A PPI network associated with migration and pyroptosis-related genes with differential expression was assembled on the basis of a minimum interaction score of 0.400, which corresponds to a medium confidence level. The Cytoscape [32] platform was employed for the visualisation of the PPI network model. It is hypothesised that regions of the PPI network that are localised and demonstrate a high degree of interconnection correspond to molecular ensembles with well-defined biological functions.

In addition, we applied cytoHubba [33] to obtain the TOP10 hub genes associated with cell migration and pyroptosis using 2 algorithms in the plugin. The 2 algorithms are Maximum Neighborhood Component (MNC) and Edge permeability component (EPC) [34]. In the PPI Network, the scores of genes that were differentially expressed in the context of cell migration and pyroptosis were calculated and subsequently sequenced in accordance with the aforementioned scores.

## 2.7 Construction of the regulatory network

Transcription factors (TFs) regulate gene expressions in the post-transcription phase by engaging with target genes (mRNA). By ChIPBase database [35] to analyze the Regulatory role of transcription factors on hub genes associated with and pyroptosis, and visualize the mRNA-TF Regulatory Network via the Cytoscape software.

It has been demonstrated that microRNAs play a pivotal role in regulating a range of biological processes, including development and evolutionary change. A single miRNA may regulate numerous target genes; alternatively, a single gene may be regulated by multiple miRNAs. The Starbase database [36] was employed for the purpose of analysing the relationship between phenotypic-related hub genes and microRNAs. The miRNAs associated with hub genes associated with cell migration and pyroptosis were obtained and intersected. Subsequently, the mRNA-miRNA Regulatory Network was rendered in a visual representation by means of the Cytoscape software.

The DGIDB (Drug-Gene Interaction Database) (https://dgidb.genome.wustl.edu/) [37] can be employed to predict direct as well as indirect drug targets of hub genes associated with cell migration and pyroptosis. In addition, the database allows researchers to explore interactions between these genes and drugs. Once this process is complete, the mRNA-Drugs Regulatory Network is constructed by means of Cytoscape software.

## 2.8 Immunoinfiltration analysis

Single-Sample Gene Set Enrichment Analysis (SSGSEA) [38], also known as single-sample Gene Set Enrichment Analysis, is a method used to quantify the relative abundance of infiltrates in each immune cell. Firstly, each distinct immune cell type was identified and labelled. These included, but were not limited to: Activated CD8 T cell, Activated dendritic cell, Gamma delta T cell, Natural killer cell, Regulatory T cell, and other human immune cell subtypes. Subsequently, the enrichment fraction, calculated by ssGSEA, was employed to represent the relative abundance of each immune cell infiltration within each sample. Samples with a p-value less than 0.05 were filtered out to obtain the immune cell infiltration matrix. Additionally, the R-package pheatmap was utilized to generate a correlation heatmap, illustrating the correlation analysis results of hub genes associated with cell migration and pyroptosis, and ssGSEA immune cells.

## 2.9 Statistical analysis

The dataset was processed through a series of calculations and statistical analyses, utilising the open-source software package R (https://www.r-project.org/, version 4.2.0). In order to compare two sets of continuous variables, the statistical relevance of variables that exhibited a normal distribution was estimated through the independent Student's t-test. Conversely, the differences between variables with non-normal distributions were analysed through the Mann-Whitney U test, otherwise known as the Wilcoxon rank sum test. All statistical tests were conducted with a level of statistical precision ($p < 0.05$) and with bilateral p-values.

# 3 Results

## 3.1 Analysis flow chart

The analytical path we followed in this study is shown in the Analysis flow chart (Fig 1).

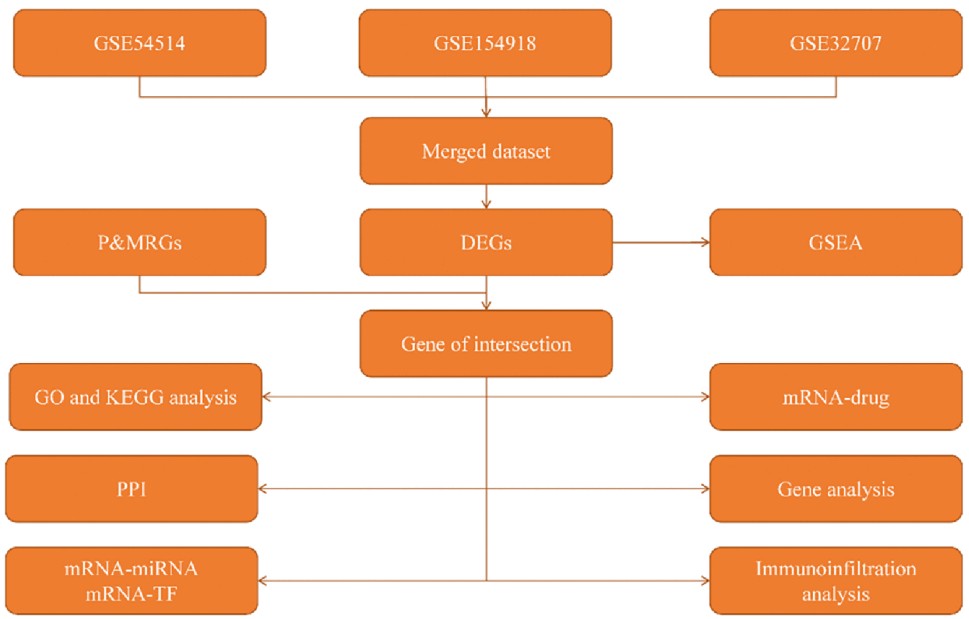

**Fig 1. Analysis flow chart.** P&MRGs: Pyroptosis & Migration-related differentially expressed genes, DEGs: Differential expression genes, GSEA: Gene Set Enrichment Analysis, GO: Gene Ontology, KEGG: Kyoto Encyclopedia of Genes and Genomes, PPI: Gene set Enrichment Analysis, PPI: protein-protein interaction network, TF: transcription factor.

### 3.2 Data sets preprocessing

We downloaded the ischaemia-reperfusion expression profile data GSE154918, GSE32707, and GSE54514 from GEO, using the normalize Between Arrays function in the limma package to normalize the data sets, along with the R package sva the Combat function in the Limma package performed a batch correction merge of the three expression data. Through normalization, we can see a similar distribution of gene expression values in the three datasets (Fig 2A and 2B), and the batch effect of the three datasets is corrected (Fig 2C and 2D), allowing for subsequent analysis.

### 3.3 Analysis of differentially expressed genes associated with sepsis

The combined data set was divided into the sepsis group and the control group. In order to analyse the discrepancy in gene expression values between the sepsis group and the control group in the combined data set, the differential expression genes of the two groups of data were obtained using the R package limma. The results were as follows: Merge data sets, a total of 3566 meet | logFC |>0 and p.adjust<0 threshold of differential expressed genes (DEGs); Under the threshold, we raised a total of 1715 expressed genes, cut a total of 1851 expressed genes, in order to get the differentially expressed genes which were connected with cell pyroptosis and migration, we get all the differentially expressed genes (DEGs) and pyroptosis & migration-related genes in intersection, a total of 23 DEGs related to pyroptosis & migration and mapping Venn (Fig 3A). The 23 pyroptosis and migration-related differentially expressed genes were as follows: *PTGS2, SIRT1, ELAVL1, ABL1, MAPK14, DHX9, TRIM24, ETS1, HMGB1, IKBKE, MST1, SQSTM1, PTPN11, ANXA1, TLR2, AGER, IL1B, PAK2, CASP3, CD274, ICAM1, CASP8, IL18*. According to the intersection results, the positions of 23 differentially expressed genes related to pyroptosis and migration on human chromosomes were analyzed by R package RCircos, and chromosomal localization maps were drawn (Fig 3B). The chromosomal mapping showed that there were two genes in chromosome 2 (*IL1B, CASP8*), two in chromosome 4 (*TLR2, CASP3*), and two (*AGER, MAPK14*) in chromosome 6; there were chromosome 1 (*PTGS2*), chromosome 9 (*CD274*), chromosome 10 (*SIRT1*), chromosome 11

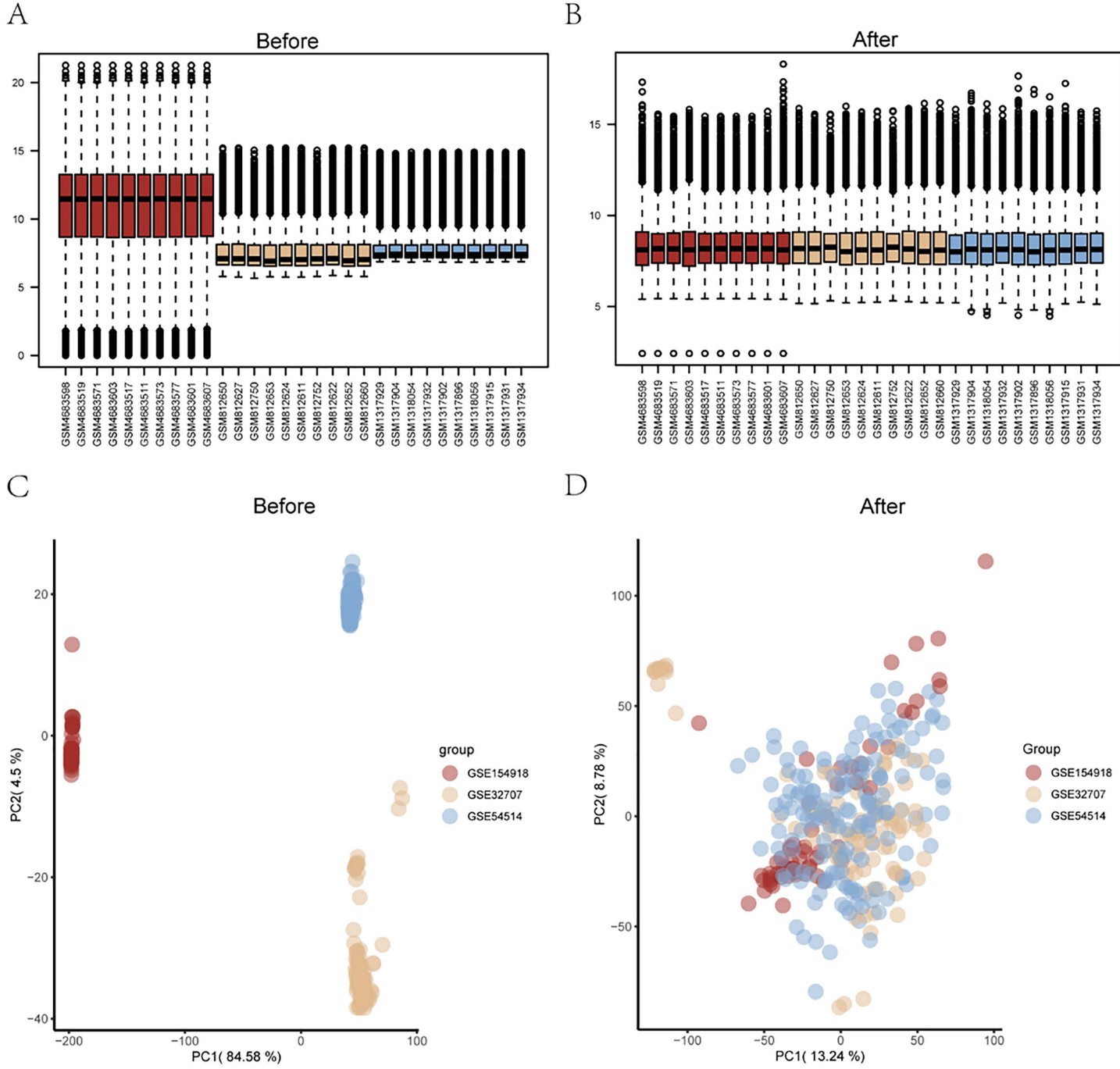

**Fig 2. Distribution of gene expression before and after batch correction in the data sets GSE154918, GSE32707 and GSE54514. (A)** Boxplot distribution of gene expression of GSE154918, GSE32707 and GSE54514 before batch correction; **(B)** Box-like distribution of gene expression of GSE154918, GSE32707 and GSE54514 after batch correction; **(C)** Boxplot distribution of gene expression of GSE154918 before batch correction; PCA analysis of GSE32707 and GSE54514, **(D)** PCA analysis of GSE154918, GSE32707 and GSE54514 after batch correction.PCA: Principal Component Analysis.

**Fig 3. Analysis of sepsis related differentially expressed genes in the combined data sets GSE154918, GSE32707 and GSE54514.** (A) Venn diagram of differential genes and pyroptosis and migration genes in the combined data set of sepsis disease group and control group. Red represents the differential gene set, and yellow represents pyroptosis and migration-related genes. (B) Location map of differentially expressed pyroptosis and migration chromosomes. (C) Differential analysis volcano map of the combined data set of sepsis disease group and control group. Red nodes represent up-regulated differentially expressed genes, blue nodes represent down-regulated differentially expressed genes, gray nodes represent genes that are not significantly differentially expressed, and labeled genes are differentially expressed pyroptosis and migration genes.(D) Heat map of gene expression in the combined data set for sepsis and control groups.The horizontal coordinate is sampled, the vertical coordinate is differentially expressed genes, red represents high gene expression, blue represents low gene expression, the red comment bar represents control group, yellow comment bar represents disease group.: Sepsis, P&MRGs: Pyroptosis & Migration-related differentially expressed genes, DEGs: Differential expression genes.

(*IL18*), chromosome 13 (*HMGB1*) and chromosome 19 (*ICAM1*). Finally, differences in the expression of pyroptosis and migration-related DEGs between different sample groups in the combined dataset were analyzed. The outcomes were illustrated with the aid of heat maps and volcano maps (see Fig 3C and 3D).

### 3.4 Gene ontology (GO) and pathway (KEGG) enrichment analysis of pyroptosis and migration related differentially expressed genes

Through GO and KEGG enrichment analysis, the relationship between biological pathways (Pathway), cell components (CC), biological processes (BP), and molecular functions (MF) of 23 pyroptosis and migration-related differentially expressed genes and sepsis was further explored. The 23 pyroptosis and migration-related differentially expressed genes were as follows: *PTGS2, SIRT1, ELAVL1, ABL1, MAPK14, DHX9, TRIM24, ETS1, HMGB1, IKBKE, MST1, SQSTM1, PTPN11, ANXA1, TLR2, AGER, IL1B, PAK2, CASP3, CD274, ICAM1, CASP8, IL18*. The 23 genes that exhibited differential expression with regard to pyroptosis and migration were subjected to GO and KEGG enrichment analysis. The particular outcomes are set forth in (S2 and S3 Tables). The results indicate that the genes associated with pyroptosis and migration are predominantly implicated in the modulation of cell-cell adhesion, the positive regulation of cytokine synthesis, the control of leukocyte cell-cell adhesion, and other BP. Furthermore, they are involved in the organization of cell components (e.g., membrane microdomains, membrane regions, and membrane rafts) and in the regulation of molecular functions (e.g., cell adhesion molecule binding, receptor tyrosine kinase binding, and mitogen-activated protein kinase binding). Additionally, it is enriched in biological pathways, including the interleukin-17 signaling pathway, the tumor necrosis factor-α pathway, and the Toll-like receptor signaling pathway. The results of GO and KEGG enrichment analysis were visualized using a bar graph (Fig 4A). Meanwhile, according to the enrichment analysis of GO and KEGG, the gene expression cycle diagram of BP, CC, and MF was drawn (Fig 4B). The gene expression string diagram pertaining to the biological pathway (Fig 4C) and the pertinent molecule and annotations for the pertinent entries were illustrated via a line description.

Additionally, the R package Pathview was employed to construct the pathway map, depicted in (Fig 4D-4F). This map serves to illustrate the primary enrichment pathways evident in the KEGG enrichment results. We showed the Toll-like receptor signaling pathway (Fig 4D), the TNF signaling pathway (Fig 4E), and the IL-17 signaling pathway (Fig 4F). At the same time, we will list the reported pathways and their related literature, clearly showing the overlap between the pathways identified in previous studies and the pathways in our study to enhance the validity of our findings. (S1 Table)

In order to ascertain the influence of gene expression levels upon sepsis, we undertook an analysis of the biological pathways impacted by gene expression within the integrated data set comprising both control and sepsis groups (see Fig 5A). Results showed that the biocarta caspase pathway (Fig 5D), il18 signaling pathway (Fig 5E), signaling by rho gtpases (Fig 5F), nod like receptor mapk6, mapk4 signaling(Fig 5G), nod like receptor signaling pathway (Fig 5H), and other biologically related pathways were up-regulated in the sepsis group. However, pi3kakt signaling pathway (Fig 5B), fischer direct p53 targets meta analysis (Fig 5C), and other biological pathways were down-regulated in the sepsis group (S4 Table).

### 3.5 Build a PPI interaction network

First, a protein-protein interaction analysis was conducted to construct a PPI Network of 23 differentially expressed genes related to pyroptosis and migration using the STRING database and visualization of the interactions using Cytoscape software (Fig 6A). The results of the PPI showed that 22 pyroptosis and migration-related differentially expressed genes were related, namely: *PTGS2, SIRT1, ELAVL1, ABL1, MAPK14, DHX9, ETS1, HMGB1, IKBKE, MST1, SQSTM1, PTPN11, ANXA1, TLR2, AGER, IL1B, PAK2, CASP3, CD274, ICAM1, CASP8, IL18*(Fig 6B), with Node Score Cutoff＝2, K-core＝2, Max.Depth＝100 as the threshold value, cluster the protein interaction network using MCODE plug-in and identify a key module: *TLR2, SIRT1, PTGS2, MAPK14, IL1B, IL18, ICAM1, HMGB1, CD274, CASP8, CASP3, AGER*. According to two

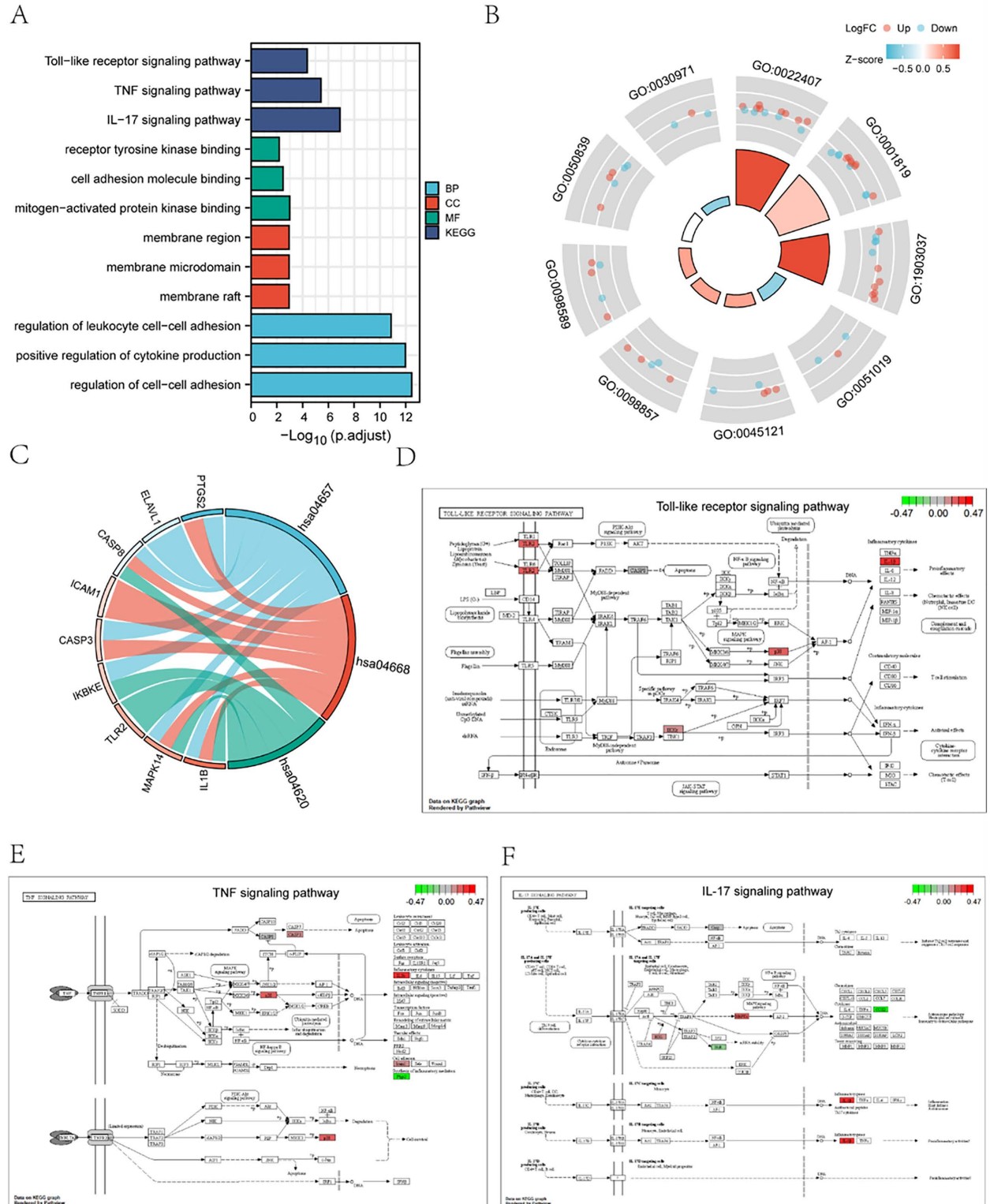

**Fig 4. Functional enrichment analysis. (A)** For GO and KEGG functional enrichment analysis, the horizontal coordinate is various GO items enriched by differentially expressed mitochondrial autophagy-related genes, and the vertical coordinate is - log10(p-value). Red represents biological processes, green represents cell components, dark blue represents molecular functions, and light blue represents enriched biological pathways. **(B)** GO enrichment

pathway circle diagram. The graph can be divided into two parts: inner circle and outer circle. Each column of the inner circle corresponds to an entry. The height of the inner circle is the relative size of p.adj. The higher the height, the smaller the p.adjust of the ID. The color of the column corresponding to fil represents the Z-score value corresponding to the entry. **(C)** KEGG enrichment analysis string diagram, which can be divided into left and right parts: The left part is the gene color block, the different colors of the color block represent the corresponding logFC value, the right part is the entry color block, the size of the color block represents the corresponding Counts (that is, the number of molecules in this entry in this enrichment analysis), the line between the left and right color blocks (the string line) represents the molecules in this entry, And the lines represent the molecules in this entry. **(D)** Toll-like receptor signaling pathway, **(E)** TNF signaling pathway, **(F)** IL-17 signaling pathway visualization results. Each node represents a gene that plays an important role in that pathway, and the node's color is determined by log2FC, with green representing differentially down-regulated genes and red representing differentially up-regulated genes.

Cytohubba plug-in algorithms, the scores of pyroptosis and migration-related differentially expressed genes were calculated, and 22 differentially expressed genes were ranked according to the scores. The two algorithms are EPC and MNC, respectively. The genes obtained by the two algorithms were the same and overlapped with the gene set obtained by MCODE. The top ten genes obtained by the two algorithms were: *IL1B, HMGB1, CASP3, IL18, SIRT1, MAPK14, CASP8, ICAM1, PTGS2, TLR2*(Fig 6C and 6D). The core gene cluster obtained by MCODE was used as hub genes for subsequent analysis.

### 3.6  Construction of mRNA-miRNA, mRNA-TF, and mRNA-drug interaction networks

The ENCORI database was utilized to obtain mRNA-miRNA data, which was then employed to predict the miRNAs interacting with 12 hub genes (mRNAs). Subsequently, the mRNA-miRNA interaction network was rendered through Cytoscape software mapping (Fig 7A). The mRNA-miRNA interaction network comprises eight hub genes (mRNAs) (*CASP3, CASP8, CD274, HMGB1, IL1B, MAPK14, PTGS2,* and *SIRT1*) and 114 miRNA molecules, as illustrated in Fig 7A. In total, 179 pairs of mRNA-miRNA interactions were identified.

We searched the CHIPBase database (version 3.0) for transcription factors (TF) that bind to hub genes. The interactions identified in the database were downloaded and the resulting data set comprised the interaction relationship between 11 hub genes (*AGER, CASP3, CASP8, CD274, HMGB1, ICAM1, IL18, IL1B, MAPK14, PTGS2, SIRT1*) and 67 TFS. And visualized via Cytoscape software (Fig 7B).

In order to determine whether there is a correlation between specific drugs or molecular compounds and the identified hub genes (mRNA), the DGIdb database was utilized. The process yielded 12 candidate drugs or molecular compounds, including those related to the following genes: *TLR2, SIRT1, PTGS2, MAPK14, IL1B, IL18, ICAM1, HMGB1, CD274, CASP8, CASP3*, and *AGER*. As illustrated in the mRNA-drug interaction network depicted in Fig 7C, we were able to identify 56 potential pharmaceutical agents or molecular compounds corresponding to eight hub genes.

### 3.7  Validation of key genes using external datasets

To further investigate the expression differences of pyroptosis- and migration-related hub genes in the combined dataset, a grouped comparison plot (Fig 8A) was used to display the differential expression analysis results of the 12 pyroptosis- and migration-related hub genes between the disease group and the control group in the combined dataset. The results indicated that the expression levels of the 12 pyroptosis- and migration-related hub genes were statistically significant (p-value $< 0.05$) between the disease and control groups. Subsequently, receiver operating characteristic (ROC) curves for the 12 pyroptosis- and migration-related hub genes in the combined dataset were plotted and presented (Fig 8B–8M). The ROC curves demonstrated that the pyroptosis- and migration-related hub genes *AGER* (AUC = 0.615, Fig 8B), *CASP3* (AUC = 0.605, Fig 8C), *CASP8* (AUC = 0.603, Fig 8D), *CD274* (AUC = 0.578, Fig 8E), *HMGB1* (AUC = 0.653, Fig 8F), *ICAM1* (AUC = 0.585, Fig 8G), *IL1B* (AUC = 0.615, Fig 8H), *IL18* (AUC = 0.633, Fig 8I), *MAPK14* (AUC = 0.684, Fig 8J), and *TLR2* (AUC = 0.629, Fig 8M) exhibited low accuracy (0.5 < AUC < 0.7) in distinguishing between the groups based on their expression levels in the combined dataset. In contrast, *PTGS2* (AUC = 0.751, Fig 8K) and *SIRT1* (AUC = 0.733, Fig 8L) showed moderate accuracy (0.7 < AUC) in differentiating the groups.

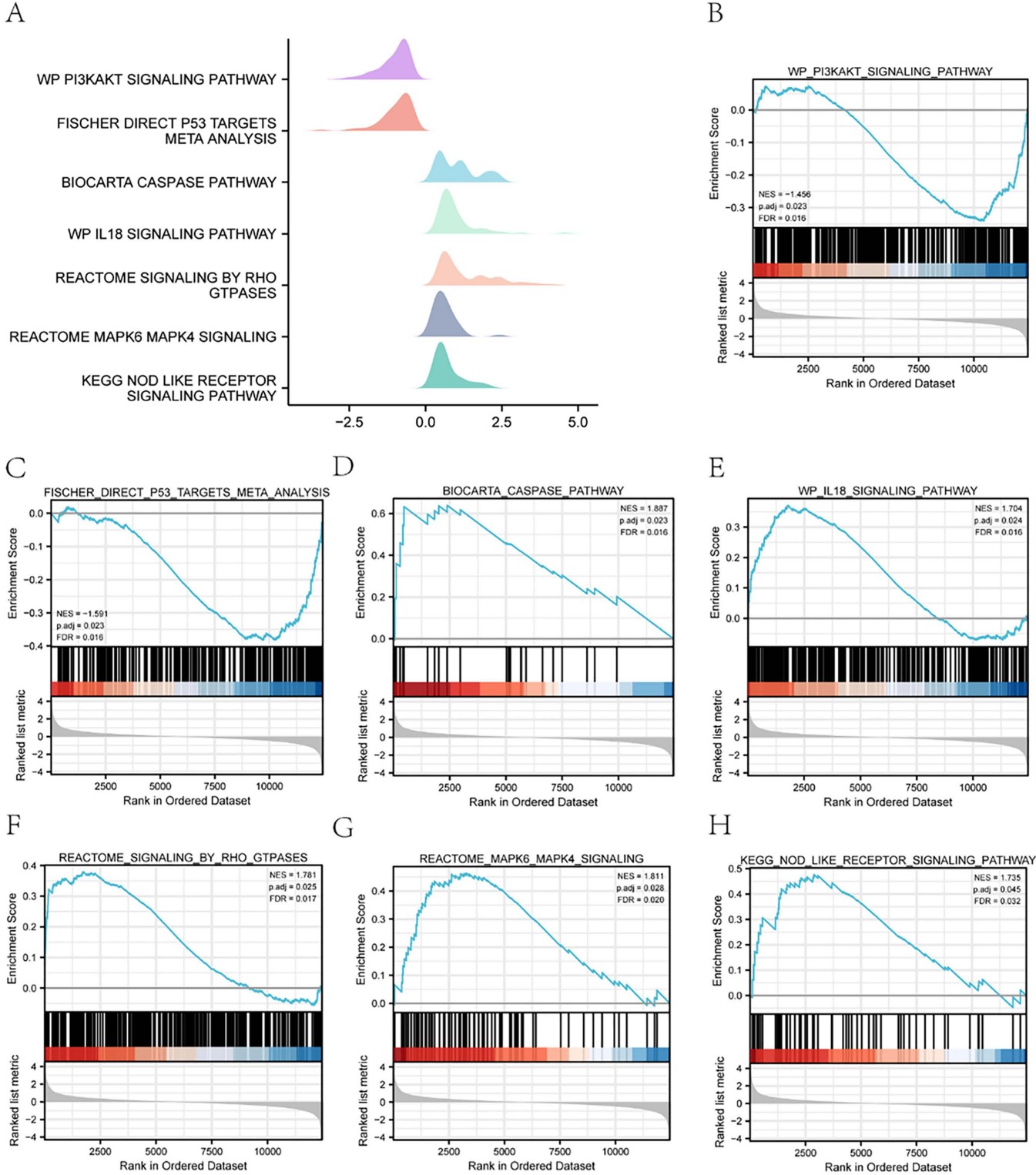

**Fig 5. GSEA. (A)** Mountainous plot of GSEA results for sepsis and control groups. **(B-H)** GSEA visualization, X-axis is the rank of genes in the list of differentially expressed genes, up-regulated is greater than zero, down-regulated is less than zero, upper Y-axis is enrichment fraction, lower Y-axis is logFC value, each color represents a pathway. Among them, biocarta caspase pathway **(D)**, il18 signaling pathway **(E)**, signaling by rho gtpases **(F)**, nod

like receptor mapk6 mapk4 signaling **(G)**, nod like receptor signaling pathway (H) was upregulated in the sepsis group.While pi3kakt signaling pathway **(B)**, fischer direct p53 targets meta analysis (C) was down-regulated in the sepsis group. GSEA: Gene Set Enrichment Analysis.

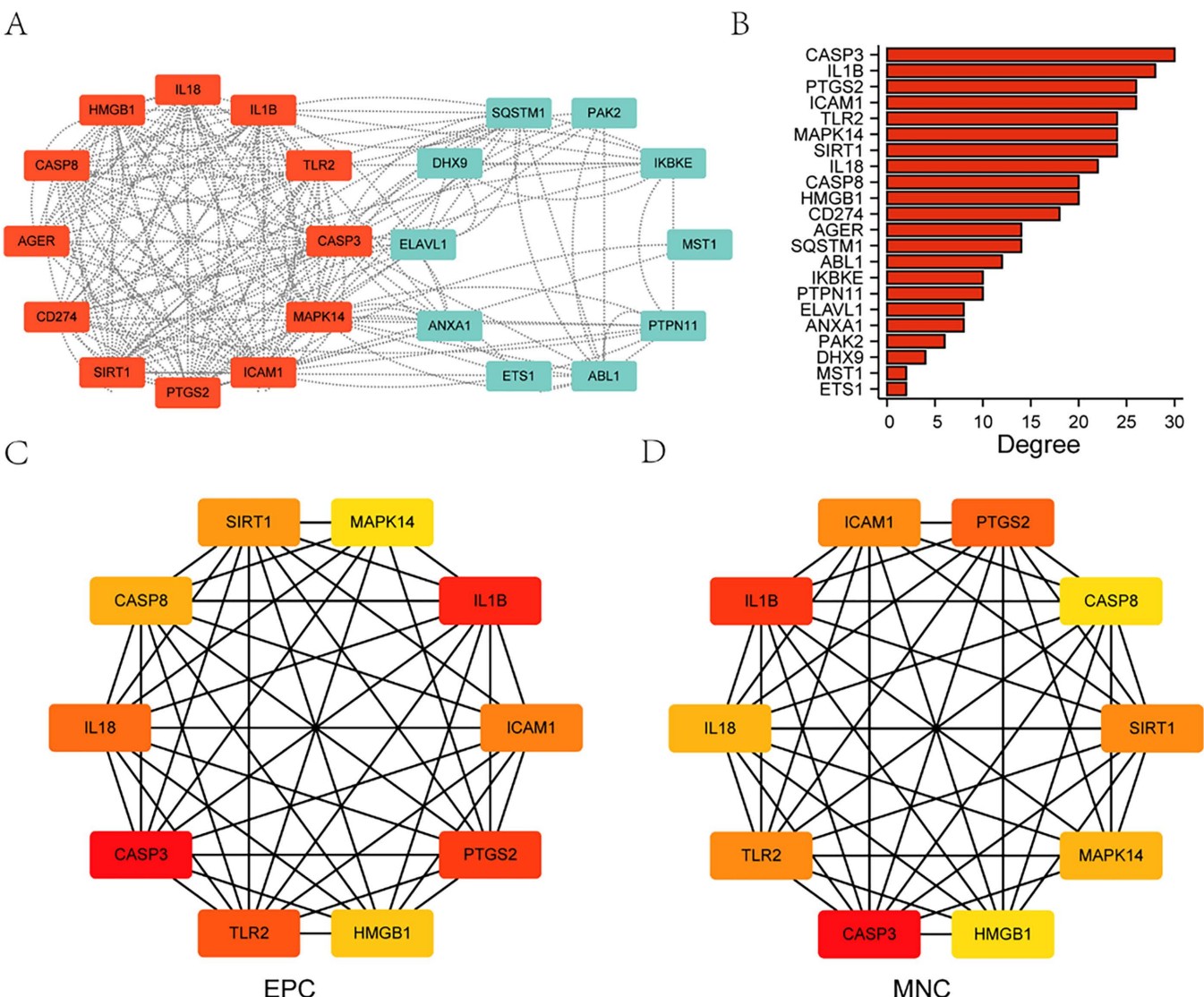

**Fig 6. Construction of PPI interaction network. (A)** 23 differentially expressed genes related to pyroptosis and migration were constructed to make a PPI network diagram, where red represents important modules of MCODE recognition. **(B)** The degree of connectivity of each node in the protein interaction network, with the x-coordinate representing the number of connections and the y-coordinate representing genes. **(C)** For the top 10 genes identified by EPC, the darker the color, the higher the score. **(D)** For the top 10 genes identified by MNC, the darker the color, the higher the rating.

## 3.8 Analysis of immunoinfiltration of hub genes related to pyroptosis and migration

To analyze the biological associations between pyroptosis- and migration-related hub genes and the immune microenvironment in sepsis, we assessed the correlations between these hub genes and immune infiltrating cells. The enrichment levels of immune cells in the samples were estimated using the ssGSEA algorithm [39]. Subsequently, correlations

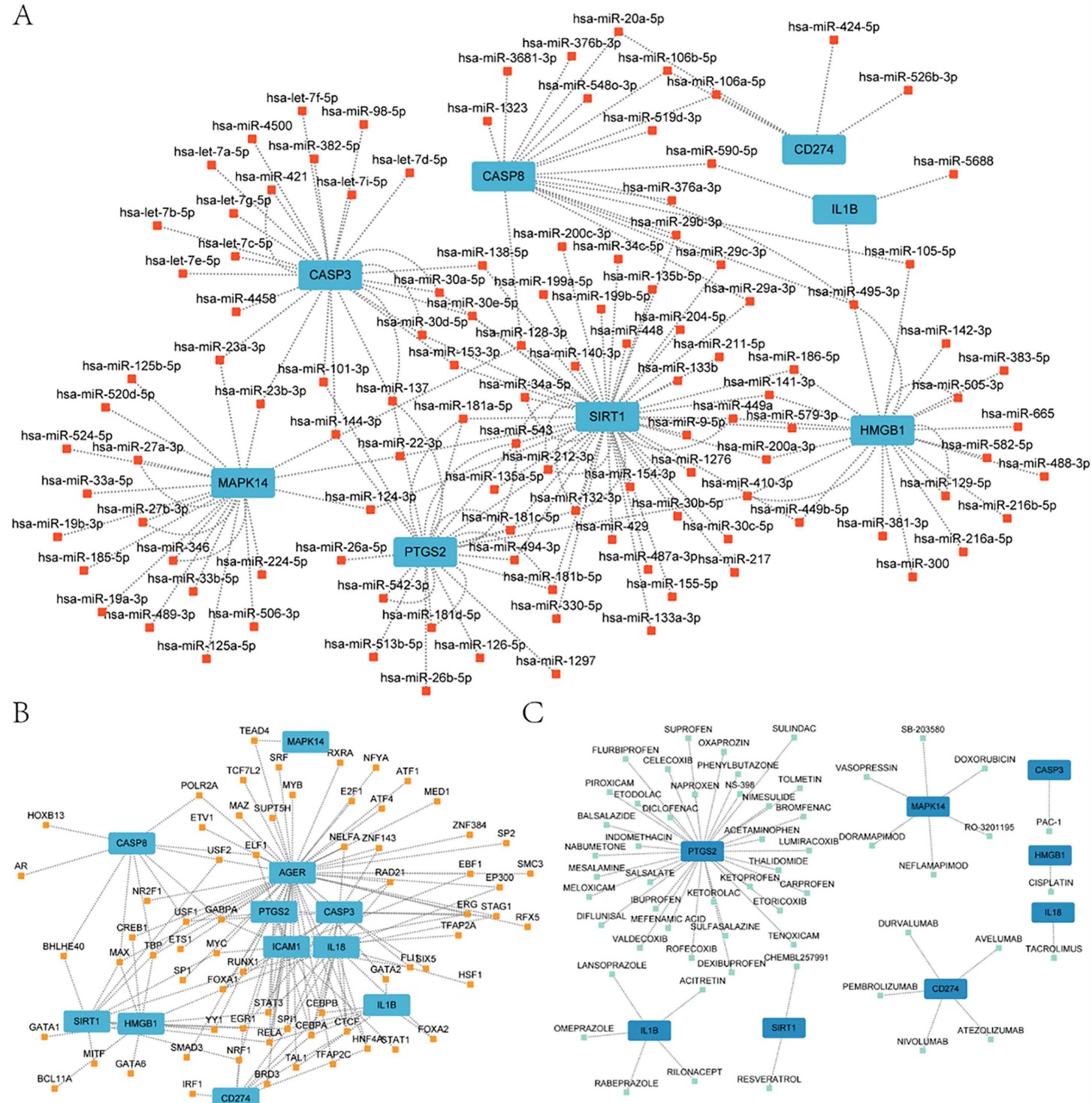

**Fig 7. Construction of mRNA-miRNA, mRNA-TF, and mRNA-drug interaction networks. (A)** Diagram of the mRNA-miRNA interaction network, where blue represents mRNA and red represents miRNA. **(B)** miRNA-TF interaction network diagram, where blue represents mRNA and yellow represents TF (Transcription Factor). **(C)** miRNA- Drug interaction network diagram, where blue represents mRNA and light blue represents drug.TF: Transcription Factor.

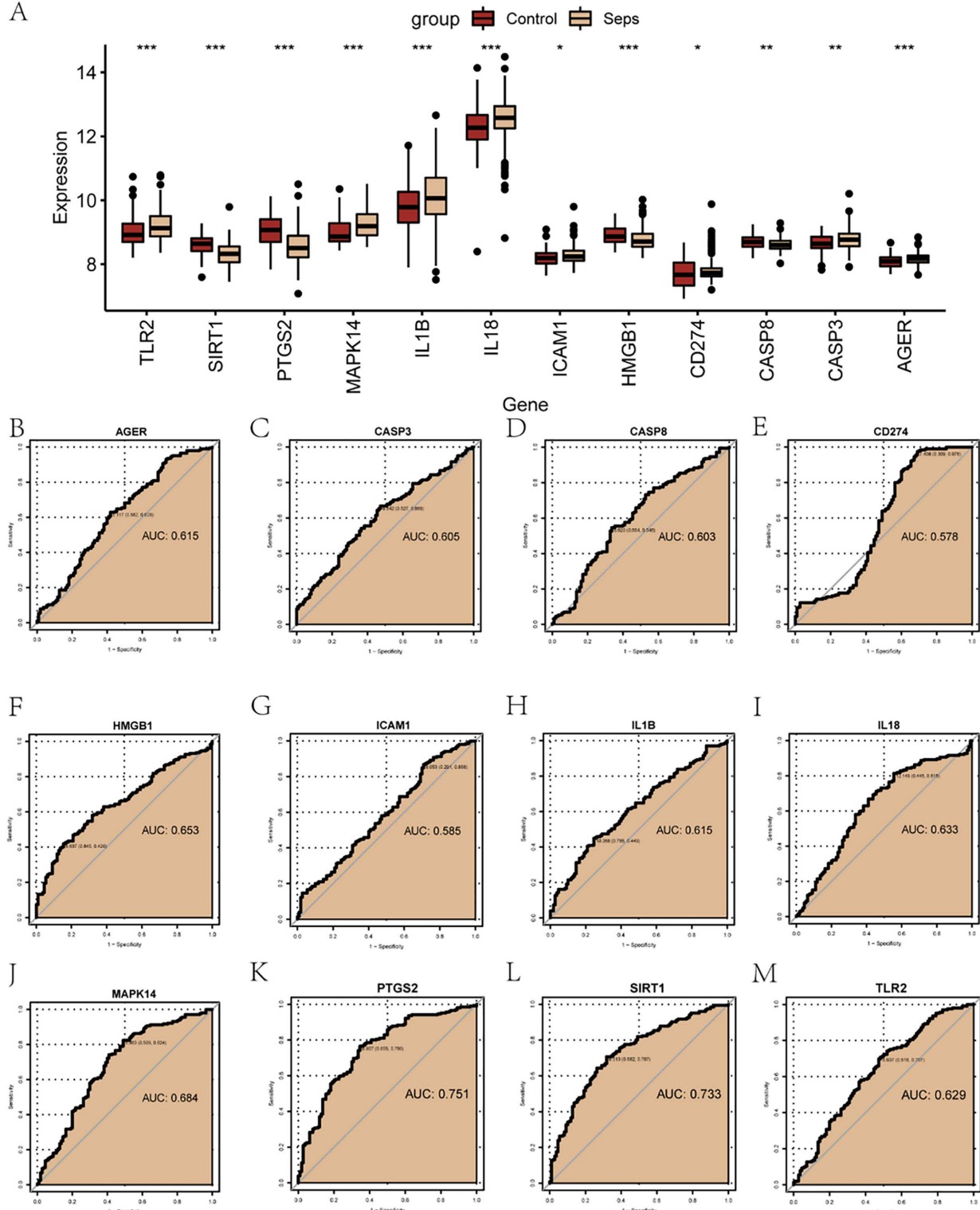

**Fig 8. Analysis of key genes. (A)** Group comparison of hub genes associated with pyroptosis and migration in the combined dataset. Red represents the control group, and yellow represents the sepsis group. * indicates p-value < 0.05, denoting statistical significance; ** indicates p-value < 0.01, denoting high statistical significance. B–M. ROC curves for *AGER* **(B)**, *CASP3* **(C)**, *CASP8* **(D)**, *CD274* **(E)**, *HMGB1* **(F)**, *ICAM1* **(G)**, *IL1B* **(H)**, *IL18* **(I)**,

MAPK14 **(J)**, *PTGS2* **(K)**, *SIRT1* **(L)**, and *TLR2* **(M)**. An area under the curve (AUC) between 0.5 and 0.7 indicates low accuracy, an AUC between 0.7 and 0.9 indicates moderate accuracy, and an AUC above 0.9 indicates high accuracy. Seps: sepsis.

between key genes and immune cells were calculated using the Pearson method and visualized via a correlation heatmap (Fig 9A). For gene–cell pairs with more pronounced correlations, scatter plots were generated to illustrate the relationships. Significant positive correlations were observed between Activated dendritic cells and *MAPK14* (R = 0.69, p < 0.01; Fig 9B), as well as between *Neutrophils* and *IL1B* (R = 0.68, p < 0.01; Fig 9C), *MAPK14* (R = 0.66, p < 0.01; Fig 9D), and *TLR2* (R = 0.65, p < 0.01; Fig 9E). In contrast, significant negative correlations were identified between *MAPK14* and Memory B cells (R = −0.50, p < 0.01; Fig 9F), Activated CD8 T cells (R = −0.46, p < 0.01; Fig 9G), and CD56dim natural killer cells (R = −0.42, p < 0.01; Fig 9H), as well as between *SIRT1* and Type 17 T helper cells (R = −0.41, p < 0.01; Fig 9I).

## 4. Discussion

Studies have shown that PCT levels increase in response to bacterial infections, particularly sepsis; however, its peak concentration is reached after a relatively prolonged period, resulting in a time delay that limits its utility for early diagnosis and immediate therapeutic decision-making. Moreover, PCT may not elevate or show significant changes in localized infections, certain Gram-positive bacterial infections, and fungal infections, thereby reducing its diagnostic sensitivity. Additionally, PCT exhibits limited specificity in distinguishing between sepsis and non-infectious systemic inflammatory response syndrome, necessitating integration with clinical features and dynamic monitoring [40,41]. CRP, as an acute-phase reactant, demonstrates high sensitivity but poor specificity. It becomes elevated in almost all inflammatory conditions, including viral infections, non-infectious inflammatory states, and postoperative responses, making it difficult to effectively differentiate between infectious and non-infectious inflammation. Furthermore, CRP has a relatively long half-life, and its serum levels lag behind actual clinical changes; thus, it cannot promptly and accurately reflect treatment efficacy or prognosis [42–45]. The clinical phenotypes of septic patients are highly diverse, encompassing a wide range of multidimensional clinical and biological features. These not only reflect the severity and stage of the disease but are also closely related to factors such as the patient's genetic background, immune status, and environmental exposures [45]. In sepsis research, existing literature primarily focuses on the regulatory roles of miRNAs in immune response [46], cell migration [47], and apoptosis [48]. For instance, miR-21 and miR-146a influence immune cell priming and apoptosis by negatively regulating the expression of inflammatory cytokines [48,49]. Furthermore, the dual effects of cell migration and pyroptosis have not been thoroughly explored in previous studies, leaving our understanding of the overall mechanisms of sepsis-associated miRNAs limited.

This study identified 23 differentially expressed genes related to pyroptosis and cell migration in sepsis. Among these, eight genes—*TLR2, SIRT1, PTGS2, MAPK14, IL18, ICAM1, CD274*, and *CASP3*—were consistently validated in an external cohort and demonstrated high discriminatory power through ROC analysis. Research indicates that TLR2, a Toll-like receptor that recognizes pathogen-associated molecular patterns (PAMPs), activates the NF-κB pathway in sepsis, promoting the release of inflammatory cytokines and exacerbating systemic inflammatory response. In septic patients, the expression of TLR2 and TLR4 on neutrophils exhibits more dynamic changes than on monocytes [50]. During sepsis, activation of TLR4 in hepatocytes upregulates caspase-11 expression, thereby promoting hepatocyte pyroptosis and exosome release [51,52]. Additionally, TLR2 signaling facilitates leukocyte migration and endothelial activation by increasing the expression of adhesion molecules, promoting immune cell infiltration [53]. SIRT1, an NAD + -dependent deacetylase, exerts anti-inflammatory and cytoprotective effects. In sepsis, SIRT1 inhibits excessive inflammatory responses by deacetylating key proteins in the NF-κB and MAPK pathways. It also regulates autophagy, mitigating organ damage caused by sepsis, and influences endothelial cell function, improving microcirculatory disorders [54,55]. PTGS2, a key enzyme in prostaglandin synthesis, is significantly upregulated in sepsis, promoting the production of the inflammatory

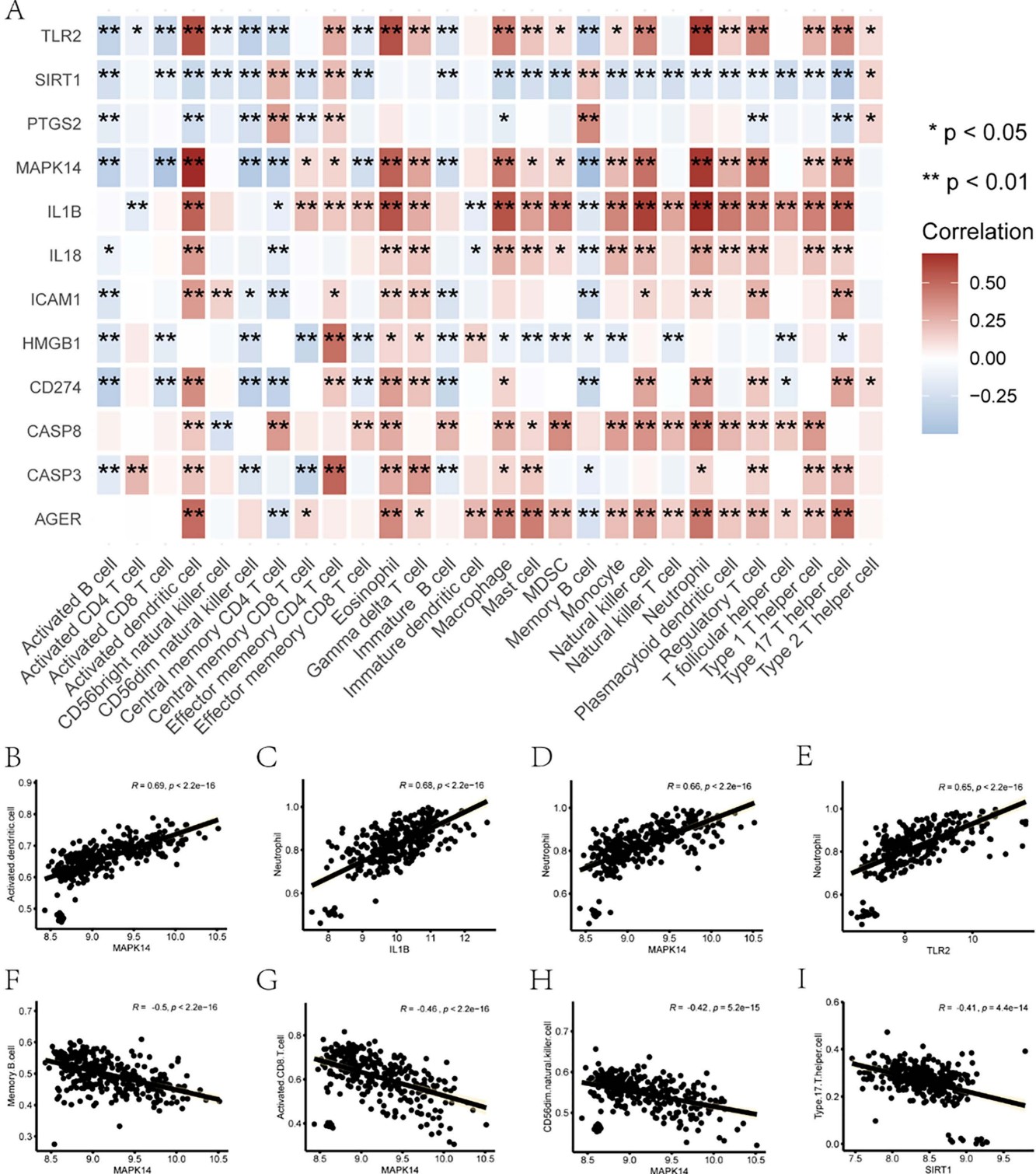

**Fig 9. Correlation of immune infiltration. (A)** Heat map showing the correlation between pyroptosis-migration-related hub genes and immunoinfil-trated cells in sepsis. The horizontal axis represents different immune infiltrating cells, and the vertical axis represents the pyroptosis and migration-related hub genes in sepsis. The redder the square color indicates the stronger the correlation between genes and immune cells. * indicates statistically

significant differences (p < 0.05), and ** indicates p < 0.01. B-I: Activated dendritic cell with *MAPK14* **(B)**, Neutrophil with *IL1B* **(C)**, *MAPK14* **(D)** and *TLR2* **(E)**, *MAPK14* with Memory B cell **(F)**, Correlation scatter plots of Activated CD8 T cell **(G)** and CD56dim natural kil er cell **(H)**, *SIRT1* and Type 17 T helper cell **(I)**.

mediator PGE2. Coupling of PGE2 receptors to different G proteins can induce diverse immune responses. For example, via EP2/EP4 receptors on dendritic cells, PGE2 stimulates interleukin-23 production, thereby inducing T helper 17 cell differentiation, while via EP3 receptors, it can activate mast cells. Leveraging these mechanisms, PGE2 plays a role in the pathogenesis of immune-related diseases [56]. MAPK14, also known as p38α, is a member of the mitogen-activated protein kinase family and serves as a profound regulator of inflammatory responses and cellular stress [57]. Studies have shown that MAPK14 exacerbates immunosuppression by enhancing neutrophil infiltration [58]. IL-18 is an inflammatory cytokine that induces IFNγ production [59]. Previous research has highlighted its critical role in sepsis, identifying it as a potential therapeutic target [60]. ICAM-1, a member of the immunoglobulin superfamily, is reported to be an important adhesion molecule [61]. Its expression is primarily found on endothelial and epithelial cells, with lower levels on specific leukocyte subsets [62]. Moreover, ICAM-1 interacts with ligands CD11/CD18 on the surfaces of neutrophils and lymphocytes, participating in neutrophil chemotaxis. The expression of ICAM-1 on neutrophils can increase inducible nitric oxide synthase (iNOS) production and neutrophil extracellular trap (NET) formation, potentially exacerbating inflammatory responses and tissue damage during sepsis [63]. CD274, also known as PD-L1, is a ligand that binds to the receptor PD1, typically on T cells, and inhibits T cell priming [64]. In a mouse model of sepsis, upregulation of PD-L1 delays neutrophil apoptosis and promotes lung injury [65].

Subsequently, the ssGSEA algorithm was employed to determine the gene expression levels of immune cells in the samples, and the Pearson method was applied to calculate the correlations between key genes and immune cells. Gene–cell pairs with the most significant correlations were selected to generate scatter plots, including activated dendritic cells with MAPK14, neutrophils with IL1B, and MAPK14 with TLR2, all of which exhibited significant positive correlations. In contrast, significant negative correlations were observed between MAPK14 and memory B cells, activated CD8+ T cells and CD56dim natural killer cells, as well as SIRT1 and T helper 17 cells. unctional enrichment analysis revealed that these genes are involved in key biological processes such as leukocyte adhesion and cytokine production. The key genes associated with pyroptosis and migration were significantly enriched in classic inflammatory signaling pathways like IL-17 and TNF. These pathways not only act as bridges in regulating inflammatory cytokine storms, immune cell recruitment, cell migration, and pyroptosis but also form the molecular basis of immune dysregulation and multi-organ damage in sepsis. The IL-17 and TNF signaling axis, as a core component of the inflammatory response, along with its upstream and downstream signal transduction networks, provides critical theoretical support for untangling the synergistic regulation of pyroptosis and migration mechanisms [66]. Further integrated bioinformatics analyses, including PPI, mRNA–miRNA, mRNA–transcription factor, and mRNA–drug networks, elucidated the multi-level regulatory mechanisms of these hub genes and identified several potential therapeutic agents.

Notably, the findings of this study are consistent with and expand upon previous research. For example, TLR2 and MAPK14, as core molecules in innate immune signaling and stressful responses, are widely involved in inflammatory injury processes when dysregulated. SIRT1 plays a protective role in organ damage such as sepsis-induced acute kidney injury by activating autophagy, further supporting its importance as a key regulatory node. Similarly, the upregulation of PTGS2 and its modulation by natural compounds like acacetin highlight its therapeutic potential. IL-18, a potent inducer of interferon-gamma, along with ICAM1, underscores the interactions between inflammation and cell migration. Additionally, CD274 and CASP3 reflect pathways of immune exhaustion and apoptosis, respectively, both of which are significant markers of sepsis progression.

Despite these strong correlations, this study has several limitations. The analysis was primarily based on whole-blood transcriptome data, which may not fully capture tissue-specific regulatory mechanisms in major organs affected by sepsis, such as the lungs and kidneys. Although external validation enhanced the reliability of the hub genes, inherent heterogeneity among public datasets—including differences in sequencing platforms, patient demographics, and sepsis etiology— may affect the generalizability of the results. Moreover, bioinformatics predictions, particularly those involving miRNA interactions and drug applicability, require further validation through ex vivo and in vivo experiments. Immune cell infiltration analysis revealed significant correlations between key genes and immune cell subtypes. For instance, MAPK14 strongly positively correlated with activated dendritic cells and neutrophils but negatively with memory B cells and cytotoxic T cells, suggesting its potential role in regulating innate and adaptive immune responses. These findings further emphasize the importance of evaluating sepsis within the context of immune cells and provide a basis for developing personalized immunotherapeutic strategies.

In summary, this integrated analysis provides new insights into the synergistic roles of cell migration and pyroptosis in the pathophysiology of sepsis and identifies eight hub genes as promising candidates for biomarker development and therapeutic target screening. Future studies should focus on functional validation of these genes, exploration of their tissue-specific roles, and evaluation of their value in sepsis subtyping and prognosis prediction.

## 5. Conclusion

Through analysis of multiple sepsis expression datasets, this study identified 23 differentially expressed genes associated with pyroptosis and migration, which were significantly altered in sepsis patients and reflect key pathological changes. Functional enrichment revealed their involvement in critical processes including intercellular adhesion, cytokine production, and leukocyte interaction, underscoring the importance of pyroptosis and migration in sepsis development. Protein-protein interaction network analysis highlighted several hub genes, such as CASP3 and TLR2, which may play central roles in regulating pyroptosis and migration and represent potential therapeutic targets. Integration of miRNA and transcription factor regulatory networks further elucidated the molecular interactions underlying sepsis pathogenesis. These hub genes show significant clinical potential as biomarkers for predicting severity and prognosis, and as targets for intervention-for example, targeting CASP3 may offer novel treatment avenues. Additionally, drug-target information based on these genes could guide personalized treatment strategies and improve clinical management. Overall, this study offers new insights into sepsis mechanisms, though further experimental validation is needed to clarify the specific roles and clinical applicability of these genes.

## Supporting information

**S1 Table. 83 genes related to cell migration and pyroptosis were obtained after intersection.**
(CSV)

**S2 Table  GO analysis of differentially expressed genes related to pyroptosis and migration: Based on the GO database.**
(DOCX)

**S3 Table.  KEGG analysis of differentially expressed genes related to pyroptosis and migration in cells: based on KEGG database.**
(DOCX)

**S4 Table.  GSEA analysis of Table4 sepsis and control group: Based on c2.cp.kegg.v7.4.entrez.gmt data set.**
(DOCX)

## Author contributions

**Conceptualization:** Ma Xu, Jinshuai Lu.

**Data curation:** Ma Xu.

**Formal analysis:** Ma Xu.

**Funding acquisition:** Jinshuai Lu.

**Investigation:** Ma Xu, Kun Han.

**Methodology:** Ma Xu, Jianhao Wang.

**Project administration:** Jianhao Wang.

**Resources:** Jinshuai Lu.

**Software:** Ma Xu, Jianhao Wang, Kun Han.

**Validation:** Jianhao Wang.

**Writing – original draft:** Ma Xu.

**Writing – review & editing:** Jinshuai Lu.

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
