## [Decision Letter · Decision Letter 0]

29 Aug 2025

Dear Dr. Xu,

Thank you for submitting your manuscript to PLOS ONE. After careful consideration, we feel that it has merit but does not fully meet PLOS ONE’s publication criteria as it currently stands. Therefore, we invite you to submit a revised version of the manuscript that addresses the points raised during the review process.

We look forward to receiving your revised manuscript.

Kind regards,

Zhanzhan Li

Academic Editor

PLOS ONE

Journal Requirements: 

3. Please note that PLOS One has specific guidelines on code sharing for submissions in which author-generated code underpins the findings in the manuscript. In these cases, we expect all author-generated code to be made available without restrictions upon publication of the work. Please review our guidelines at https://journals.plos.org/plosone/s/materials-and-software-sharing#loc-sharing-code and ensure that your code is shared in a way that follows best practice and facilitates reproducibility and reuse.

Additional Editor Comments:

Reviewer #1:

Reviewer #2:

Reviewers' comments:

Reviewer's Responses to Questions

**Comments to the Author**

1. Is the manuscript technically sound, and do the data support the conclusions?

Reviewer #1: Yes

Reviewer #2: Yes

2. Has the statistical analysis been performed appropriately and rigorously?

Reviewer #1: Yes

Reviewer #2: Yes

3. Have the authors made all data underlying the findings in their manuscript fully available?

Reviewer #1: Yes

Reviewer #2: Yes

4. Is the manuscript presented in an intelligible fashion and written in standard English?

Reviewer #1: Yes

Reviewer #2: Yes

Reviewer #1: 1. Strengths

The bioinformatics pipeline (e.g., batch correction with sva, DEG analysis with limma, enrichment with clusterProfiler) is standard and well-executed, with external validation adding credibility.

Results are logically presented, and the identification of 8 validated hub genes (TLR2, SIRT1, PTGS2, MAPK14, IL18, ICAM1, CD274, CASP3) is supported by ROC curves (AUC >0.7).

Data availability is fully met, and no funding/competing interests are declared.

-Major Concerns and Suggestions

Methods and Results Soundness:

The methods are detailed and reproducible, but some choices (e.g., lenient DEG threshold of logFC >0) could introduce noise. Conclusions are supported, but interpretations occasionally overreach (e.g., miRNA's "critical role" without direct evidence).

Discussion Section Suggestions:

The Discussion is comprehensive, linking results to sepsis biology and acknowledging limitations (e.g., dataset heterogeneity, lack of wet-lab validation). However, it could be more concise (reduce repetition of intro material) and focused on interpreting results within the study's scope. Strengthen by:

Explicitly tying hub genes to sepsis mechanisms without overgeneralizing (e.g., specify that findings are from blood-derived data, limiting tissue-specific inferences).

Expanding on implications for bioinformatics in sepsis research, emphasizing how this integrates pyroptosis/migration (supported by your enrichments in IL-17/TNF pathways).

Discussing whether the 8 hub genes are novel in the sepsis context or already known in multiple inflammatory settings (e.g., ICAM1 and CD274 are well-established in endothelial activation and immune checkpoint roles across infections and cancers, as per Meng et al., 2016, Genet Mol Res, and Masugi et al., 2017, Gut).

Clarifying if the study suggests causal mechanisms (e.g., via pathway enrichments) or merely associations (e.g., correlations in immune infiltration), to avoid implying untested causality.

Further Analyses to Enhance Robustness (Not Currently Performed):

To bolster the analysis, consider these additions, which are common in bioinformatics studies and would improve confidence in results without requiring new data:

Weighted Gene Co-expression Network Analysis (WGCNA): Identify co-expression modules among the 23 DEGs and correlate with sepsis traits (e.g., severity). This could reveal if hub genes cluster together, strengthening PPI findings (e.g., as in Li et al., 2023, Front Immunol on pyroptosis-related genes).

Single-Cell RNA-Seq Integration: Map bulk DEGs to public scRNA-seq sepsis datasets (e.g., GSE167363) to resolve cell-type specificity (e.g., pyroptosis in neutrophils vs. macrophages).

Survival Analysis: If metadata allows (e.g., from GSE57065), perform Kaplan-Meier on high/low hub gene expression for mortality prediction.

Alternative Cutoff Testing: Perform and report DEG analysis with stricter logFC cutoffs (e.g., |logFC| >0.5 or >1), as low thresholds (|logFC|>0) can inflate gene lists and FDR.

Validation Across Subgroups: Where data allows, repeat analyses on subsets (e.g., adult vs. pediatric, early vs. late sepsis, infection source) to assess robustness of hub gene selection across patient subtypes.

Methodological Comparison: Conduct pathway enrichment with alternative tools (enrichR, g:Profiler) to confirm core findings are not tool-dependent.

Network Analysis Robustness: Re-run hub selection using multiple network centrality metrics (Degree, Closeness, Betweenness) and show overlap with the presented results.

Sensitivity analyses test result stability under varied parameters. Suggested ones:

DEG Threshold Variation: Re-run with logFC >0.5 or >1 to assess if the 23 DEGs/hubs change (common to check robustness, e.g., in Gao et al., 2024, Gene).

Bootstrap Resampling for Enrichments: Bootstrap GO/KEGG (1000 iterations) to compute confidence intervals for p-values, reducing false positives from small gene sets.

PPI Interaction Score Variation: Test STRING scores >0.7 (high confidence) vs. 0.4 to evaluate hub stability.

Cross-Validation in ssGSEA: Split data into training/test sets for immune correlations to avoid overestimation.

Partial Rank Correlation or Variance-Based Sensitivity Analyses: Apply in gene/pathway prioritization steps to check for consistent ranking under parameter perturbations.

How were quality control (QC) checks performed before and after normalization? Were outlier samples removed?

Were PCA plots quantitatively assessed (variance explained by PC1/PC2 before vs. after correction) rather than visually?

Was log2 transformation applied consistently?

Which immune cell signature set was used for ssGSEA—LM22 (CIBERSORT) or another? Was it validated for whole blood transcriptomics?

Were multiple testing corrections applied to the gene–cell correlation analyses?

Could strong positive/negative correlations be due to overall neutrophil or lymphocyte proportions rather than functional associations?

Reviewer #2: Comments to the Author

I appreciate the opportunity to review this manuscript, which addresses a highly relevant clinical issue: the identification of key genes related to sepsis, cell migration, and pyroptosis through bioinformatics analysis. The study is well-structured and represents a potentially valuable contribution to the understanding of sepsis pathophysiology. Below I provide my observations.

Strengths

Clear and specific title, focused on the central aim of the study.

The introduction adequately contextualizes the clinical relevance of sepsis and the need for new biomarkers.

The importance of miRNAs, cell migration, and pyroptosis in the pathogenesis of sepsis is well justified.

The bioinformatics methodology is robust, integrating multiple datasets and employing enrichment analyses, PPI networks, and external validation.

Results are presented in a logical and progressive manner:

Identification of 3,566 DEGs.

Selection of 23 genes related to pyroptosis/migration.

Validation of 8 key genes (TLR2, SIRT1, PTGS2, MAPK14, IL18, ICAM1, CD274, and CASP3).

Relevant biological pathways (IL-17, TNF, TLR) are identified, and immune infiltration analysis highlights meaningful correlations.

The conclusion appropriately proposes that the validated genes may represent potential biomarkers and therapeutic targets.

Overall, the manuscript is clearly written, requiring only minor stylistic and grammatical adjustments for international publication.

Areas for Improvement

The main weakness is the lack of experimental validation (in vitro or in vivo), which limits the clinical translation of the findings. It is recommended to emphasize that results should be considered exploratory and require confirmation in experimental models.

There is heterogeneity across datasets (different platforms and patient characteristics). Although batch correction was applied, a more detailed discussion of the potential impact of these differences would strengthen the manuscript.

The discussion could be enhanced in two key aspects:

Clinical applications, contrasting the proposed genes with currently used biomarkers (e.g., procalcitonin, CRP).

The relationship with secondary complications of sepsis, such as cardiac dysfunction, renal injury, or lung injury.

Including a perspective on epigenetic processes (particularly DNA methylation) would enrich the analysis, as these mechanisms regulate gene expression and may affect the identified genes.

Relevant proteins involved in cell migration, such as matrix metalloproteinases (MMPs), were not considered. A brief discussion of their role would complement the analysis.

Some network figures (PPI, KEGG, mRNA-miRNA) are complex and difficult to interpret; highlighting key genes more clearly would improve readability.

The manuscript could benefit from reducing redundancies in the text (e.g., repeated gene lists in multiple sections) to improve conciseness.

Overall Recommendation

The manuscript presents a solid and well-structured bioinformatics analysis that identifies genes of interest in sepsis. However, to strengthen its scientific impact, I recommend:

Expanding the clinical and comparative discussion.

Emphasizing methodological limitations more explicitly.

Refining text, figures, and references.

With these improvements, the article could provide a significant contribution and serve as a solid foundation for future translational studies in sepsis

**Do you want your identity to be public for this peer review?** For information about this choice, including consent withdrawal, please see our Privacy Policy

Reviewer #1: **Yes: ** Ali Afkhaminia

Reviewer #2: **Yes: ** César Bermúdez-Mejía

---

## [Author Response · Author response to Decision Letter 1]

10 Oct 2025

Dear Editor / Editor and Reviewers

Manuscript ID: PONE-D-25-35332

Title: PPI-based screening of hub genes related to sepsis migration/pyroptosis and immune infiltration analysis

We are deeply grateful to you and the other reviewers for your invaluable comments and suggestions, which have played a crucial role in significantly improving the quality of our manuscript. In response to the thorough and insightful feedback provided, we have conducted extensive revisions to the paper. We have carefully addressed each of the points raised, supplemented relevant content where necessary, and refined several sections to enhance clarity and completeness. A detailed point-by-point response to all the reviewers’ comments has been prepared, and all changes in the manuscript have been highlighted in red for your convenience. We sincerely hope that the revised version meets with your approval and are more than willing to make any further adjustments as needed. Thank you very much for your time and support—it is a great honor to have our work considered for publication in PLOS ONE.

RE We sincerely appreciate your thorough review and valuable comments.

We will meticulously revise the manuscript in strict accordance with the latest PLOS ONE template, comprehensively reviewing and adjusting the manuscript structure, formatting, file naming, figure/table uploads, captions, and reference style to ensure all submitted materials fully comply with the journal's requirements.

Thank you once again for your valuable suggestions.

RE Thank you for your careful review and valuable comments. We will supplement the ORCID iD of corresponding authors in the author information section in subsequent revisions, and ensure that verification is completed in the Editorial Manager system to meet journal-related requirements.

Thank you again for your valuable advice.

3. Please note that PLOS One has specific guidelines on code sharing for submissions in which author-generated code underpins the findings in the manuscript. In these cases, we expect all author-generated code to be made available without restrictions upon publication of the work. Please review our guidelines at https://journals.plos.org/plosone/s/materials-and-software-sharing#loc-sharing-code and ensure that your code is shared in a way that follows best practice and facilitates reproducibility and reuse.

RE Thank you for your thorough review and valuable comments. All analytical procedures in this study were performed using R scripts developed by the authors. The relevant code and documentation have been prepared in accordance with the journal’s requirements. Upon publication of the manuscript, all author-generated code and associated documentation will be made publicly available on a mainstream open-source platform. Detailed access information will be provided in the “Data and Code Availability Statement” section of the subsequent manuscript to ensure the reproducibility and reusability of the research findings.

We sincerely appreciate your insightful suggestions.

RE Thank you for your thorough review and valuable feedback. We will carefully verify and standardize the funding information in the subsequent revision to ensure consistency between the "Funding Information" and "Financial Disclosure" sections.

Once again, we appreciate your insightful suggestions.

RE We sincerely appreciate your thorough review and valuable suggestions. Based on your recommendations, this study will conduct a comprehensive evaluation of the relevance of the suggested literature and will incorporate appropriate citations in subsequent revisions as needed, thereby ensuring the scientific rigor and representativeness of the references cited.

Thank you once again for your insightful comments.

Additional Editor Comments:

Reviewer #1:

Reviewer #2:

Reviewers' comments:

Reviewer's Responses to Questions

Comments to the Author

1. Is the manuscript technically sound, and do the data support the conclusions?

Reviewer #1: Yes

Reviewer #2: Yes

2. Has the statistical analysis been performed appropriately and rigorously?

Reviewer #1: Yes

Reviewer #2: Yes

3. Have the authors made all data underlying the findings in their manuscript fully available?

Reviewer #1: Yes

Reviewer #2: Yes

4. Is the manuscript presented in an intelligible fashion and written in standard English?

Reviewer #1: Yes

Reviewer #2: Yes

5. Review Comments to the Author

Reviewer #1: 1. Strengths

The bioinformatics pipeline (e.g., batch correction with sva, DEG analysis with limma, enrichment with clusterProfiler) is standard and well-executed, with external validation adding credibility.

Results are logically presented, and the identification of 8 validated hub genes (TLR2, SIRT1, PTGS2, MAPK14, IL18, ICAM1, CD274, CASP3) is supported by ROC curves (AUC >0.7).

Data availability is fully met, and no funding/competing interests are declared.

-Major Concerns and Suggestions

Methods and Results Soundness:

The methods are detailed and reproducible, but some choices (e.g., lenient DEG threshold of logFC >0) could introduce noise. Conclusions are supported, but interpretations occasionally overreach (e.g., miRNA's "critical role" without direct evidence).

RE We sincerely thank the reviewers for their careful reading and valuable comments. In the current analysis, a relatively lenient threshold of logFC > 0 was adopted, primarily due to the generally limited expression magnitude observed in sepsis blood samples. This threshold was used during the initial screening phase to ensure biological sensitivity. However, in subsequent analyses, multiple layers of filtering and validation using an independent external dataset were applied to enhance the reliability of the results and mitigate potential noise introduced by the initial screening. Regarding the role of miRNAs in sepsis, we will revise the manuscript accordingly to more cautiously emphasize the predicted significance of miRNAs as potential regulatory factors, while explicitly stating that further experimental validation is required to elucidate their specific mechanisms. Detailed modifications can be found in the Discussion section.

Once again, we appreciate your insightful suggestions.

Discussion Section Suggestions:

The Discussion is comprehensive, linking results to sepsis biology and acknowledging limitations (e.g., dataset heterogeneity, lack of wet-lab validation). However, it could be more concise (reduce repetition of intro material) and focused on interpreting results within the study's scope. Strengthen by:

RE Thank you for your thorough review and valuable comments. We will revise the Discussion section in accordance with your suggestions by reducing redundant background information, placing greater emphasis on the core findings of this study and their scientific significance, and providing a more in-depth interpretation of the relevant signaling pathways and immune mechanisms. Specific revisions are detailed in the Discussion section.

Once again, we sincerely appreciate your insightful recommendations.

Explicitly tying hub genes to sepsis mechanisms without overgeneralizing (e.g., specify that findings are from blood-derived data, limiting tissue-specific inferences).

RE Thank you for your thorough review and valuable comments. All data analyses and conclusions presented in this study are based on human peripheral blood samples. The mechanistic inferences regarding the relevant hub genes are made solely in response to the molecular characteristics observed in the peripheral blood microenvironment. We will further emphasize the tissue origin of our findings—specifically, the peripheral blood—in key sections such as the abstract, discussion, and conclusion, to avoid extrapolation to other tissues or organs. Additionally, a corresponding limitation will be explicitly addressed in the discussion section.

We once again express our gratitude for your insightful suggestions.

Expanding on implications for bioinformatics in sepsis research, emphasizing how this integrates pyroptosis/migration (supported by your enrichments in IL-17/TNF pathways).

RE Thank you for your thorough review and valuable comments. In the revised manuscript, we will further incorporate bioinformatics approaches to explore in depth the role of pyroptosis and migration-related mechanisms in sepsis, as well as their regulatory significance within key inflammatory pathways such as IL-17 and TNF. Specific revisions are detailed in the Discussion section.

Once again, we appreciate your insightful suggestions.

Discussing whether the 8 hub genes are novel in the sepsis context or already known in multiple inflammatory settings (e.g., ICAM1 and CD274 are well-established in endothelial activation and immune checkpoint roles across infections and cancers, as per Meng et al., 2016, Genet Mol Res, and Masugi et al., 2017, Gut).

RE Thank you for your thorough review and valuable comments. In accordance with your suggestions, we have further clarified the existing research background of the eight hub genes in sepsis and other inflammatory conditions in the revised manuscript, supplemented relevant literature, and highlighted the positioning and novelty of this study. Specific revisions can be found in the Discussion section.

Once again, we sincerely appreciate your insightful recommendations.

Clarifying if the study suggests causal mechanisms (e.g., via pathway enrichments) or merely associations (e.g., correlations in immune infiltration), to avoid implying untested causality.

RE We sincerely thank the reviewers for their thorough evaluation and valuable comments. In both pathway enrichment analysis and immune infiltration analysis, this study was based on statistical correlations and did not make direct inferences regarding causal mechanisms. All results involving pathways and immune cells were described using terms such as “associated” or “enriched,” avoiding any implication of causality. In the Discussion and Conclusion sections, we further explicitly stated that the findings of this study are based on correlational analyses and do not infer causation, while also emphasizing the need for future validation through functional experiments. Relevant revisions have been made accordingly in the Discussion and Conclusion sections.

Once again, we express our gratitude for the insightful suggestions.

Further Analyses to Enhance Robustness (Not Currently Performed):

To bolster the analysis, consider these additions, which are common in bioinformatics studies and would improve confidence in results without requiring new data:

Weighted Gene Co-expression Network Analysis (WGCNA): Identify co-expression modules among the 23 DEGs and correlate with sepsis traits (e.g., severity). This could reveal if hub genes cluster together, strengthening PPI findings (e.g., as in Li et al., 2023, Front Immunol on pyroptosis-related genes).

Single-Cell RNA-Seq Integration: Map bulk DEGs to public scRNA-seq sepsis datasets (e.g., GSE167363) to resolve cell-type specificity (e.g., pyroptosis in neutrophils vs. macrophages).

Survival Analysis: If metadata allows (e.g., from GSE57065), perform Kaplan-Meier on high/low hub gene expression for mortality prediction.

Alternative Cutoff Testing: Perform and report DEG analysis with stricter logFC cutoffs (e.g., |logFC| >0.5 or >1), as low thresholds (|logFC|>0) can inflate gene lists and FDR.

Validation Across Subgroups: Where data allows, repeat analyses on subsets (e.g., adult vs. pediatric, early vs. late sepsis, infection source) to assess robustness of hub gene selection across patient subtypes.

Methodological Comparison: Conduct pathway enrichment with alternative tools (enrichR, g: Profiler) to confirm core findings are not tool-dependent.

Network Analysis Robustness: Re-run hub selection using multiple network centrality metrics (Degree, Closeness, Betweenness) and show overlap with the presented results.

RE Thank you for your thorough review and valuable feedback. In the current study, we have integrated multiple transcriptomic datasets for comprehensive analysis and validated key findings using external datasets, with a focus on the association between differentially expressed genes, protein–protein interaction networks, and immune infiltration. This approach demonstrates strong systematic and scientific rigor. Due to limitations in sample size, metadata completeness, and clinical information in the available datasets, relevant analyses have not yet been conducted. However, we fully acknowledge the importance of such analyses for in-depth mechanistic studies. In the future, we plan to incorporate more extensive clinical samples and multi-omics data to further expand related analyses, thereby continuously enhancing the breadth and depth of this research.

Once again, we appreciate your insightful suggestions.

Sensitivity analyses test result stability under varied parameters. Suggested ones:

DEG Threshold Variation: Re-run with logFC >0.5 or >1 to assess if the 23 DEGs/hubs change (common to check robustness, e.g., in Gao et al., 2024, Gene).

RE Thank you for your thorough review and valuable feedback. In the analytical process of this study, a lenient threshold of logFC > 0 was adopted during the initial screening phase, primarily due to the objective characteristic of generally low expression levels of genes in sepsis blood samples. The rationale behind this setting was to maximize the retention of potential biological signals, ensure biological sensitivity in the preliminary screening res

---

## [Decision Letter · Decision Letter 1]

4 Nov 2025

PPI-based screening of hub genes related to sepsis migration/pyroptosis and immune infiltration analysis

PONE-D-25-35332R1

Dear Dr.Xu,

We’re pleased to inform you that your manuscript has been judged scientifically suitable for publication and will be formally accepted for publication once it meets all outstanding technical requirements.

Kind regards,

Zhanzhan Li

Academic Editor

PLOS ONE

Additional Editor Comments (optional):

Reviewers' comments:

Reviewer's Responses to Questions

**Comments to the Author**

Reviewer #1: All comments have been addressed

Reviewer #2: All comments have been addressed

2. Is the manuscript technically sound, and do the data support the conclusions?

Reviewer #1: Yes

Reviewer #2: Yes

3. Has the statistical analysis been performed appropriately and rigorously?

Reviewer #1: Yes

Reviewer #2: Yes

4. Have the authors made all data underlying the findings in their manuscript fully available?

Reviewer #1: Yes

Reviewer #2: Yes

5. Is the manuscript presented in an intelligible fashion and written in standard English?

Reviewer #1: Yes

Reviewer #2: Yes

Reviewer #1: This is a strong revision. The authors have demonstrated a high level of engagement with the review process and have made meaningful improvements to their manuscript. The study presents a valuable bioinformatics resource that identifies a robust set of hub genes at the intersection of pyroptosis and cell migration in sepsis, with clear insights into their potential functional roles and regulatory networks.

The responses to the reviewers are comprehensive and satisfactory. I believe the manuscript, with the revisions as outlined in the response letter, is now suitable for acceptance in PLOS ONE

Reviewer #2: All observations have been fully addressed, resulting in a manuscript that is more solid in its technical and scientific aspects.

**Do you want your identity to be public for this peer review?** For information about this choice, including consent withdrawal, please see our Privacy Policy

Reviewer #1: **Yes: ** Ali Afkhaminia

Reviewer #2: **Yes: ** César Bermúdez-Mejía

---

## [Editor Report · Acceptance letter]

PONE-D-25-35332R1

PLOS ONE

Dear Dr. Xu,

I'm pleased to inform you that your manuscript has been deemed suitable for publication in PLOS ONE. Congratulations! Your manuscript is now being handed over to our production team.

Kind regards,

on behalf of

Dr. Zhanzhan Li

Academic Editor

PLOS ONE